# Differential growth regulates asymmetric size partitioning in *Caulobacter crescentus*

Tin Wai Ng[1,2,*] , Nikola Ojkic[3,*], Diana Serbanescu[1,2], Shiladitya Banerjee[4]

Cell size regulation has been extensively studied in symmetrically dividing cells, but the mechanisms underlying the control of size asymmetry in asymmetrically dividing bacteria remain elusive. Here, we examine the control of asymmetric division in *Caulobacter crescentus*, a bacterium that produces daughter cells with distinct fates and morphologies upon division. Through comprehensive analysis of multi-generational growth and shape data, we uncover a tightly regulated cell size partitioning mechanism. We find that errors in division site positioning are promptly corrected early in the division cycle through differential growth. Our analysis reveals a negative feedback between the size of daughter cell compartments and their growth rates, wherein the larger compartment grows slower to achieve a homeostatic size partitioning ratio at division. To explain these observations, we propose a mechanistic model of differential growth, in which equal amounts of growth regulators are partitioned into daughter cell compartments of unequal sizes and maintained over time via size-independent synthesis.

## Introduction

During a single cell cycle, bacteria face the demanding task of replicating their genomes, synthesizing sufficient surface area and macromolecular content for their progeny. To ensure the viability of the next generation, bacterial cells also need to partition DNA and determine the plane of division in such a way that the daughter cells end up with morphologies best adapted to their environment. Various models for bacterial cell size control have been proposed and disputed in recent years (1, 2, 3, 4), including the *sizer* model (5)—where cells attain a critical mass before division—and the widely prevalent *adder* model, wherein cells add a fixed length during each division cycle regardless of their initial size (6, 7, 8, 9, 10). While extensive investigation has been focused on the molecular and biophysical mechanisms governing cell size regulation in symmetrically dividing cells (11, 12, 13, 14, 15, 16), the

regulatory strategies used by asymmetrically dividing bacteria to control the size ratio of daughter cells remain poorly understood (17, 18, 19, 20).

One of the most well-studied asymmetrically dividing organisms is *Caulobacter crescentus*, a Gram-negative bacterium, which produces two genetically identical but morphologically distinct daughter cells: a motile, flagellated "swarmer" cell and a sessile but replication-competent "stalked" cell that possesses an adhesive stalk on its end (Fig 1A). *C. crescentus* cells have several checkpoints in place to ensure that asymmetric development is coordinated with the cell cycle progression (22). Although most of the cell cycle checkpoints have been established from prior genetics studies (23), the morphology of the cell also provides valuable information about the cell cycle stage (17, 18, 19, 24, 25). As shown previously (17), *C. crescentus* stalked cells have a distinct invagination near the mid-cell, which is noticeable right at the beginning of the staked cell cycle and becomes more prominent as the cell cycle progresses (Fig 1A). This invagination, characterized by the minimum of the cell width (Fig 1B), partitions the cell into swarmer (*Sw*) and the stalked (*St*) compartments. Furthermore, it was shown that the position of the minimum cell width eventually becomes the location of the cell division plane (17). Therefore, the lengths of the stalked and the swarmer compartments can be precisely tracked throughout the division cycle, from birth ($t = 0$) to division ($t = \tau$), allowing us to investigate size regulation of cell compartments in an asymmetrically developing bacterium.

In this study, we combined experimental data analysis and mathematical modeling to investigate the control of asymmetric size partitioning in *C. crescentus* cells. We performed statistical analysis of single-cell growth and shape data of *C. crescentus* cells obtained using a previously developed platform that combined temperature-controlled microfluidic chamber and image analysis pipeline for extracting cell shapes (17, 26). This platform facilitated the morphologies of single stalked cells over multiple generations under balanced growth and low density conditions (26). Using this pipeline, we analyzed the morphological changes in the stalked and the swarmer cell compartments throughout the division cycle in *C. crescentus*, by measuring their relative and absolute sizes, as well

[1]Department of Physics and Astronomy, University College London, London, UK   [2]Institute for the Physics of Living Systems, University College London, London, UK   [3]School of Biological and Behavioural Sciences, Queen Mary University of London, London, UK   [4]Department of Physics, Carnegie Mellon University, Pittsburgh, PA, USA

Correspondence: shiladtb@andrew.cmu.edu
*Tin Wai Ng and Nikola Ojkic contributed equally to this work

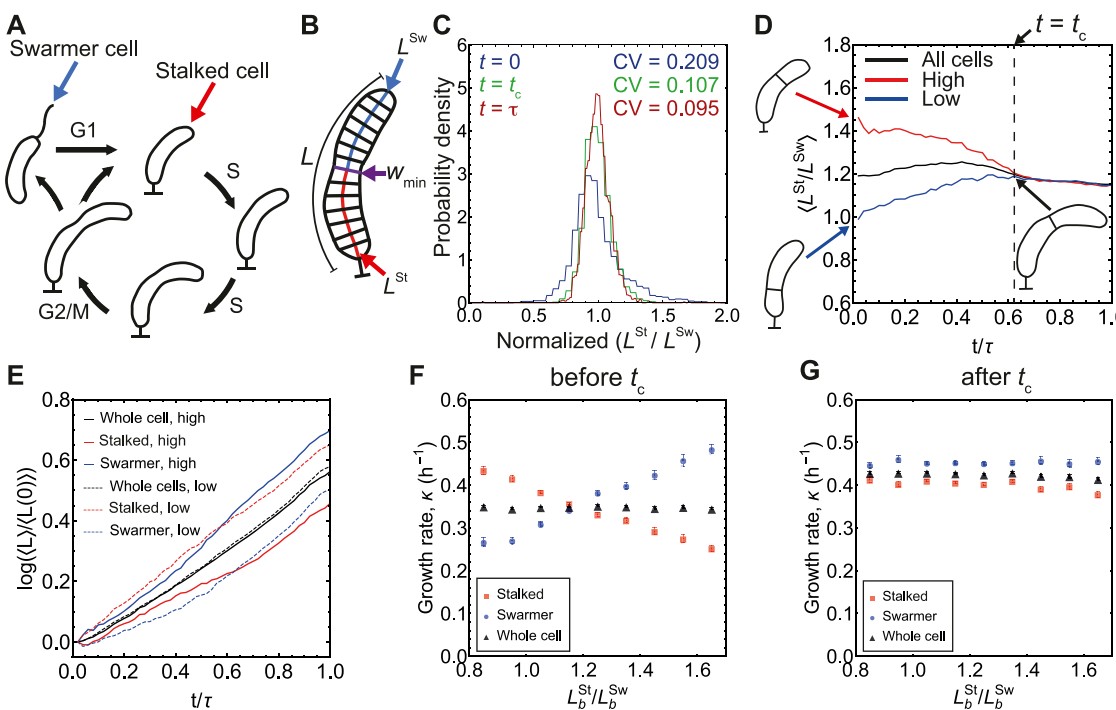

**Figure 1.   *C. crescentus* cells correct daughter cell size ratios via differential growth.**
**(A)** A schematic diagram illustrating the cell cycle of *C. crescentus*. Here we study cell size regulation during the stalked cell cycle (S/G2/M phase) (21). **(B)** Definition of cell shape variables. The minimum cell width $w_{min}$ is determined from a sample of equidistant normal lines to the midline axis. The length of the cell midline axis ($L$) from the width minimum ($w_{min}$) to the stalked pole is defined as the length of the stalked compartment ($L^{St}$), whereas the rest of the length ($L - L^{St}$) is defined as the length of the swarmer compartment ($L^{Sw}$). Both $L^{St}$ and $L^{Sw}$ can be tracked throughout the division cycle. **(C)** Probability distribution of $L^{St}/L^{Sw}$ at birth ($t = 0$), at the onset of rapid constriction ($t = t_c = 0.6\tau$), and at division ($t = \tau$), normalized by mean $L^{St}/L^{Sw} = 1.21, 1.196, 1.155$ corresponding to each time point at $t = 0$, $t = t_c$ and $t = \tau$, respectively. **(D)** Mean ratio of stalked and swarmer compartment lengths ($L^{St}/L^{Sw}$) during division cycle versus normalized time ($t/\tau$). The lines are mean $L^{St}/L^{Sw}$ that are binned in normalized time intervals of equal lengths (50 bins in total). Red line corresponds to initial length ratio $L^{St}/L^{Sw} > 1.4$ ("High"), blue line is for $L^{St}/L^{Sw} < 1.0$ ("Low"), and black line is for all cells. Crossover time $t_c$ (dashed line) marks the transition to cell constriction (18) when the length ratios reach their average value. See Fig S1 for intergenerational variability in $L^{St}/L^{Sw}$ and representative single generation trajectories. **(E)** Mean logarithm of the ratio of the stalked, swarmer and whole-cell length to their corresponding lengths at birth, taken from 50 relative time intervals of equal length in one division cycle. Solid lines represent the "high" subset ($L^{St}/L^{Sw} > 1.4$), whereas dashed lines represent the "low" subset ($L^{St}/L^{Sw} < 1.0$). **(F)** Mean growth rate before $t_c$ of stalked, swarmer and the whole cell, binned by $L_b^{St}/L_b^{Sw}$. **(G)** Mean growth rate after $t_c$ of stalked, swarmer and the whole cell, binned by $L_b^{St}/L_b^{Sw}$. Error bars represent ±1 SEM.

as their growth rates in nutrient rich medium (PYE) at different temperature conditions.

We find that correction in division plane positioning begins early in the cell division cycle through differential growth of the stalked and swarmer cell compartments. Analysis of single-cell data suggests that the bigger cell compartment grows slower to maintain a homeostatic size ratio between the daughter cell compartments. To explain these data, we developed a mathematical model for asymmetric size control in which growth regulators are partitioned in fixed amounts into unequal sized cell compartments and maintained via size-independent synthesis. This model successfully explains the maintenance of size asymmetry in daughter cell compartments and provides a mechanism for how the swarmer and the stalked cell compartments regulate growth rates to achieve the correct division ratio. In contrast to the well-established adder model for symmetrically dividing cells in which added length is independent of birth length, our model leads to a new concept of size asymmetry control via differential growth. In this paradigm, the difference in the size of the daughter cell compartments at division is maintained at a constant value independent of their birth size, establishing a new mechanism for robust control of asymmetric size partitioning.

# Results

## Differential growth corrects errors in size partitioning

To study the control of asymmetric size partitioning in *C. crescentus* cells, we first analyzed cell growth dynamics during the division cycle with various size discrepancies between the stalked and swarmer compartments. *C. crescentus* cells exhibit a pronounced invagination near the cell center, which is identifiable at the beginning of the division cycle, even before the onset of constriction (17). The location of the invagination, which ultimately becomes the division plane, divides the cell into two compartments—the stalked compartment, with length $L^{St}$, and the swarmer compartment with length $L^{Sw}$ (Fig 1B). As previously reported (17), *C. crescentus* cells exhibit a tight regulation of the division size ratio. The ratio of the stalked-to-swarmer compartment lengths at division is ≈1.2, with a coefficient of variation ($CV$) ≈ 0.095. Similar observations for high precision in division plane positioning have been reported for symmetrically dividing *Escherichia coli* and *Bacillus subtilis* with $CV$ of 0.05 and 0.08, respectively (27). However, the ratio of stalked-to-swarmer

compartment lengths ($L^{St}/L^{Sw}$) at the beginning of the division cycle is a noisy parameter with CV = 0.209 (Fig 1C), which may be attributed in part to the shallowness of the minimum width at early times, making it difficult to precisely determine the location of primary invagination. However, as the cell cycle advances, the invagination becomes more pronounced, facilitating a more accurate determination of ($L^{St}/L^{Sw}$). We asked how *C. crescentus* cells achieve a high precision in ($L^{St}/L^{Sw}$) in predivisional cells.

From a previously published dataset of 2,448 individual cell generations at 24°C (17), we extracted a subset where the stalked compartment size is substantially larger than the swarmer compartment size at birth ($L^{St}/L^{Sw}$ > 1.4, *n* = 403), and conversely a subset ($L^{St}/L^{Sw}$ < 1.0, *n* = 347) where the swarmer compartment is larger. Surprisingly, at 60% of the division cycle period τ, both the subsets reach the average value ⟨$L^{St}/L^{Sw}$⟩ ≈ 1.2 (Figs 1D and S1A–D). This time point coincides with a previously reported crossover time ($t_c$ = 0.63τ) between lateral cell-wall growth and septal growth in *C. crescentus* (18), which marks the onset of rapid cell-wall constriction. This suggests that the cell corrects deviations from the average value for $L^{St}/L^{Sw}$ before the onset of cell-wall constriction.

To investigate how individual compartments dynamically regulate their size, we computed the instantaneous growth rate κ for the whole cell using the formula, $\kappa(t) = L(t)^{-1}dL(t)/dt = d\log L(t)/dt$, and the instantaneous growth rates for the individual cell compartments as $\kappa^{St}(t) = d\log L^{St}(t)/dt$ and $\kappa^{Sw}(t) = d\log L^{Sw}(t)/dt$. With these definitions, the gradients of the curves in Fig 1E ($\log[L(t)/L(0)]$ versus *t*) indicate the instantaneous growth rates for the whole cell and its two compartments. At the beginning of the cell division cycle, the swarmer compartment in the "high" $L^{St}/L^{Sw}$ subset ($L^{St}/L^{Sw}$ > 1.4 at birth) has a higher growth rate than the stalked compartment, whereas, the swarmer compartment grows slower than the stalked compartment in the "low" $L^{St}/L^{Sw}$ subset ($L^{St}/L^{Sw}$ < 1.0 at birth). Towards the end of the division cycle, both compartments achieve similar growth rates, regardless of the initial size discrepancy (Fig 1E and G). This implies that at the beginning of the division cycle (before $t_c$), the stalked and the swarmer compartments grow at different rates to correct the deviations from the average value for $L^{St}/L^{Sw}$ (Fig 1F). The difference in their growth rates (pre-$t_c$) also increases as $L_b^{St}/L_b^{Sw}$ deviates from the range 1.1–1.2, where most cells fall into (Fig 1F). Interestingly, after $t_c$, both growth rates reach similar values, with that of the swarmer compartment being slightly higher, regardless of $L_b^{St}/L_b^{Sw}$ (Fig 1G). These correlations also hold at other temperature conditions where cells grew at different rates (Fig S2A–D). Generally, the cell compartments grow faster after $t_c$, leading to a higher whole-cell growth rate, consistent with previous measurements (18). Whereas it is known how *C. crescentus* cells globally control cell size (18), the existing models for cell size control are agnostic about how individual cell compartments regulate their size. The differential growth of the stalked and swarmer compartments implies the existence of a negative feedback between compartment size and their respective growth rates. This begs the question of how each cell compartment regulates its size during the division cycle.

## Asymmetric size control ensures asymmetric size partitioning at division

Cell size control in *C. crescentus* has recently been studied for individual stalked and swarmer cells (7, 12, 18, 25, 26). Analysis of single-cell growth and morphological data revealed that *C. crescentus* stalked cells follow the mixer model for cell size homeostasis that combines an adder and a timer component (18): $L_d = a·L_b + \Delta$, where $L_b$ is the cell length at birth (i.e., at the beginning of the stalked cell cycle), $L_d$ is the cell length at division (Fig 2C), and the parameters *a* and Δ depend on the growth conditions. This model, however, does not reveal how individual cell compartments regulate their size to ensure proper size asymmetry between the daughter cells.

At any time, $L(t) = L^{St}(t) + L^{Sw}(t)$. We find that the amount of length added in the stalked compartment during the division cycle, $\Delta L^{St}$, is negatively correlated with the length added in the swarmer compartment $\Delta L^{Sw}$ (Fig 2A). With an adder model for size control (*a* = 1), $\Delta L = \Delta L^{St} + \Delta L^{Sw}$ is a constant. Thus, we would predict a slope –1 for the correlation between $\Delta L^{St}$ and $\Delta L^{Sw}$. However, the data significantly deviate from the adder model predictions (Fig 2A). The division lengths of stalked and swarmer compartments ($L_d^{St}$ and $L_d^{Sw}$) are moderately correlated to the birth lengths of the respective compartments ($L_b^{Sw}$ and $L_b^{St}$), whereas they are strongly correlated to the total cell length at birth (Fig 2B). From multivariate least square model fit, we found

$$L_d^{St} = 0.93 + 0.63L_b^{St} + 0.59L_b^{Sw}, \quad (1)$$

$$L_d^{Sw} = 0.57 + 0.61L_b^{St} + 0.63L_b^{Sw}. \quad (2)$$

Because, $L_d = 1.49 + 1.24 L_b$ for the same set of cells (Fig 2C, top left), the regression coefficients for $L_b^{St}$ and $L_b^{Sw}$ in Equations (1) and (2) are approximately half of the regression coefficient of $L_b$. This suggests a simple mathematical model for asymmetric size control:

$$L_d^{St} \approx \frac{a}{2}L_b + \delta, \quad (3)$$

$$L_d^{Sw} \approx \frac{a}{2}L_b + \Delta-\delta, \quad (4)$$

where δ = 0.93 *μ*m, Δ = 1.49 *μ*m, *a* = 1.24. The above relations are in excellent quantitative agreement with the least square linear fit to the mean trend in the experimental data $L_d^{St}$ versus $L_b$ and $L_d^{Sw}$ versus $L_b$ (Fig 2C). This model also predicts that the difference in lengths between the stalked and the swarmer compartments at division is constant and uncorrelated to cell length at birth:

$$L_d^{St} - L_d^{Sw} \approx 2\delta - \Delta. \quad (5)$$

Indeed, experimental data show that there is no correlation between $L_d^{St} - L_d^{Sw}$ and $L_b$ (Fig 2C, bottom right). Furthermore, the mean of the binned data is in excellent agreement with the predicted value 2δ – Δ = 0.37 *μ*m, at all temperatures. Thus, the asymmetric size control models given by Equations (3) and (4) lead

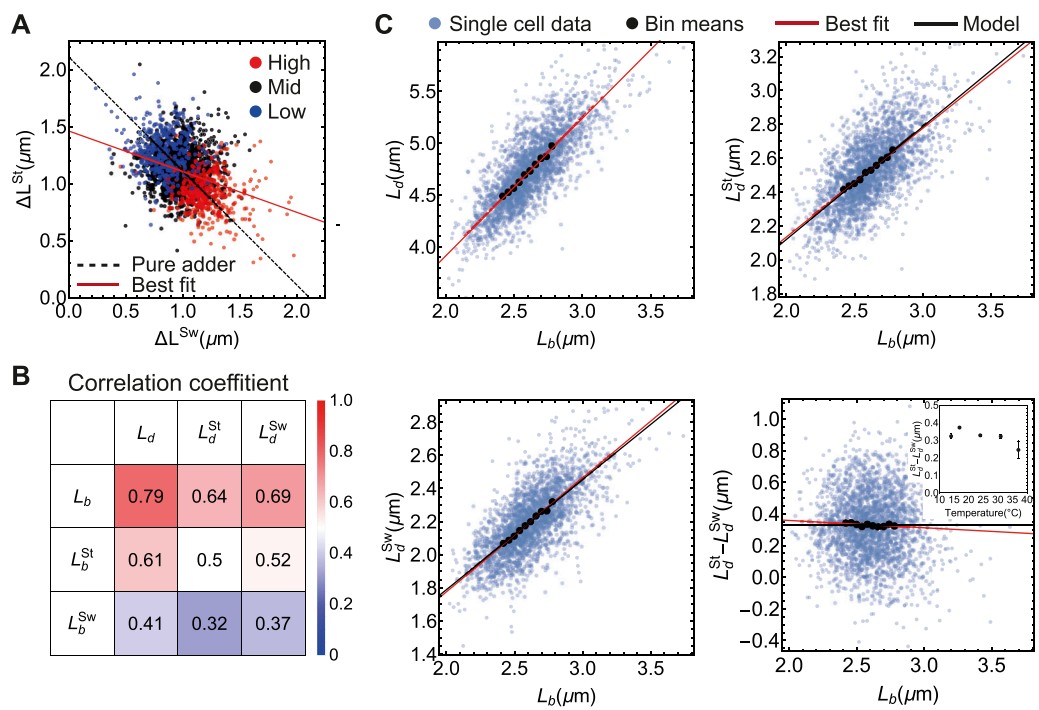

**Figure 2. Size regulation of stalked and swarmer cell compartments.**
**(A)** Added length in the stalked compartment ($\Delta L^{St}$) versus added length in the swarmer compartment ($\Delta L^{Sw}$) during one division cycle. The "Mid" subset consists of points with $1.0 \leq L_b^{St}/L_b^{Sw} \leq 1.4$. Dashed line corresponds to the fit of the adder model ($\Delta L^{St} + \Delta L^{Sw} = \langle \Delta L \rangle$, $R^2 = -0.1976$). Red solid line represents a least square linear fit to all data points ($n = 2{,}448$, $R^2 = 0.1442$). **(B)** Correlation coefficient matrix between length variables in all cells in the sample ($n = 2{,}448$). **(C)** Scatter plots of various length variables at division versus $L_b$. Red lines represent least square fits of the data corresponding to the bin means, which are taken from 10 equally spaced bins between $L_b = 2.4$ μm and $L_b = 2.8$ μm. Black line displays the expected correlation according to Equations (3), (4), and (5). (inset) Difference between the stalked and the swarmer compartment lengths at division is independent of the temperature.

to a constant size difference between the stalked and the swarmer compartments at the time of cell division.

## Size-independent partitioning of growth regulators ensures robust asymmetric size partitioning

Experimental data (Fig 1F) suggest a negative feedback between the size of individual compartments and their respective growth rates, such that the larger compartment grows at a slower rate to correct for initial size discrepancies. This differential rate of change in compartment length could arise from differential elongation rates in each compartment, or active movement of the division plane relative to the cell. In the absence of experimental evidence for the latter, we constructed a mathematical model for differential growth. In this model, we assume that the growth rate of the cell is proportional to the concentration of a regulatory molecule (which we call a *growth regulator*), whose abundance at time $t$ is given by $\varepsilon(t)$. At the beginning of the stalked cell cycle, these growth regulators can be partitioned between the stalked and swarmer compartments in two possible ways: *partitioning by size* and *partitioning by amount* (Fig 3A). In the first model, the growth regulators are partitioned such that their abundance in each compartment is proportional to the size of that compartment, resulting in equal concentration in each compartment. The growth rate, being proportional to the concentration, is the same in each compartment and thus independent of compartment size. Partitioning by size, therefore, cannot account for size-dependent

growth such that the bigger compartment grows slower to achieve the correct size partitioning ratio at division (Fig 1F).

By contrast, if the growth regulators are partitioned in fixed amounts between the two compartments, then the regulator concentration is lower in the bigger compartment compared to the smaller compartment (Fig 3A). As a result, we would expect a negative correlation between growth rate and the compartment size, consistent with experimental data.

While our model does not explicitly identify the growth regulators, we can narrow down the potential candidates based on the physical model. To maintain differential concentration, it is necessary that these growth regulators do not significantly diffuse through the cytoplasm since diffusion would tend to equalize concentration. Furthermore, diffusion barrier in *C. crescentus* is set just before cell division (28). Therefore, one possible hypothesis is that the growth regulators are immobile by being bound to the genome, which is partitioned evenly between the stalked and the swarmer cells and DNA replication begins early in the S-phase of the cell cycle (29). Possible candidates for these growth regulators include RNA polymerases, mRNAs and actively translating ribosomes, which are known to regulate bacterial growth rate (30, 31, 32, 33), and display limited mobility owing to their localization at the sites of transcription (34). Experimental observations indicate minimal mobility of active ribosomes in *C. crescentus*, characterized by a low-micrometer-scale diffusion coefficient (34, 35). This limited diffusivity can be attributed to the

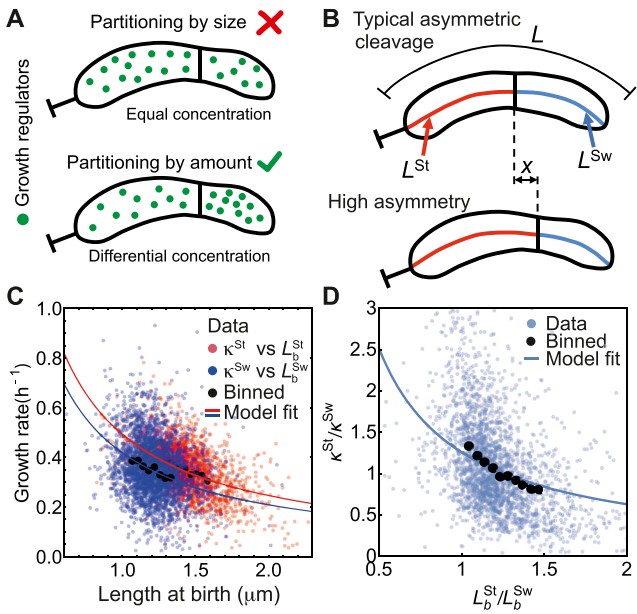

**Figure 3. Cell size correction requires size-independent partitioning of molecules regulating growth.**
**(A)** Schematic of model for size-based partitioning (top) and amount-based partitioning (bottom) of growth regulators. Amount-based partitioning model ensures differential concentration of growth regulators, thereby predicting a negative correlation between growth rates and lengths of each compartment. Solid lines near the mid-cell do not represent a physical barrier but the location of the minimum cell width separating the swarmer and stalked compartments. **(B)** Schematic of differential growth model for typical asymmetric and highly asymmetric size ratios. **(C)** Amount-based partitioning model predicts negative correlation between individual compartment growth rates and birth lengths (solid lines). Scattered points are experimental data, black points are binned data, and lines are model fit to (8). **(D)** The model predicts negative correlation between growth rate ratios and birth length ratios, given by Equation (14). Best fit to binned data, $\kappa^{St}/\kappa^{Sw} = (1.26 \pm 0.03)L_b^{Sw}/L_b^{St}$.

localization translating ribosomes at mRNA transcription sites, anchored to their corresponding genes (34, 36). Thus, the chromosome can serve as an internal template for amount-based partitioning of growth regulators into daughter cells, much like chromatin-based partitioning of cell size regulators in budding yeast and plant cells (20, 37). Aside from ribosomes, differential recruitment of MreB and PBP2 could also play a role in differential growth of cell compartments.

### Mathematical model for asymmetric growth control
We first start with a mathematical model for whole-cell growth, where the rate of cell elongation is proportional to the abundance of the regulators $\varepsilon$,

$$\frac{dL}{dt} = \alpha \int_0^L \lambda(l)dl = \alpha\varepsilon(t), \tag{6}$$

where $\alpha$ is a constant that depends on cell geometric parameters and the speed of peptidoglycan insertion, and $\lambda$ is the number density of the regulatory molecules that coordinate cell growth. If $\lambda$ has a specific spatial profile such that the growth is localized to a fixed region on the cell surface (2, 19, 38, 39, 40, 41), then $\varepsilon = \int_0^L \lambda(l)dl$ is constant and independent of $L$. This results in a negative

feedback between cell length and growth rate $\kappa = \alpha\varepsilon/L$, consistent with data for *M. smegmatis* cells where growth is localized to the cell's poles (42) (Fig S3A). By contrast, *E. coli* cells exhibit uniform lateral growth along the cell, such that $\varepsilon \propto L$. As a result, there is negligible correlation between growth rate and cell length (Fig S3B). On the other hand, *C. crescentus* cells grow by inserting peptidoglycan both laterally and at the septal plane (18, 38, 44). *C. crescentus* cell growth is uniform before the beginning of the constriction phase ($t < t_c$), whereas growth is localized to the septum after $t > t_c$ (18). Our model then predicts no correlation between cell length at birth and the growth rate before $t_c$, and a negative correlation between cell length at $t_c$ and growth rate after $t_c$, which is consistent with experimental data (Fig S3C).

We apply the whole-cell growth model to each compartment of the cell, assuming that the rate of change in length of the stalked and the swarmer compartments are given by

$$\frac{dL^i}{dt} = \alpha\varepsilon^i(t), \tag{7}$$

where $i = \{St, Sw\}$ and $\varepsilon^i$ is the abundance of the growth regulators in compartment $i$. Instantaneous growth rate of each compartment is then given by

$$\kappa^i(t) = \frac{1}{L^i(t)}\frac{dL^i}{dt} = \frac{1}{L^i(t)}\alpha\varepsilon^i(t). \tag{8}$$

If the growth regulators are partitioned in proportion to compartment size then $\varepsilon^i \propto L^i$, leading to a constant size-independent growth rate in each compartment, inconsistent with experimental data. By contrast, if the growth regulators are partitioned by amounts independent of size, then $\varepsilon^i$ does not dependent on $L^i$. It then follows from Equation (8) that $\kappa^i \propto 1/L^i$. To quantify the relationship between growth rate and compartment length, we note that the cell compartments grow at equal rates if the stalked-to-swarmer length ratio at birth is equal to its average value (Figs 1F and G and 3A). Therefore, $\kappa_b^{St} = \kappa_b^{Sw}$ if $L_b^{St} = \langle L_b^{St} \rangle$ and $L_b^{Sw} = \langle L_b^{Sw} \rangle$, where the subscript (b) refers to values at birth at the beginning of the stalked cell cycle and the angular brackets denote average across all cells. This results in the constraint

$$\frac{\varepsilon_b^{St}}{\varepsilon_b^{Sw}} = \frac{\langle L_b^{St} \rangle}{\langle L_b^{Sw} \rangle} = \frac{0.55}{0.45} = \gamma^*, \tag{9}$$

where $\gamma^*$ is defined as the average size ratio between stalked and swarmer cell compartments at birth. The numerical value for $\gamma^*$ is determined by the size control parameters $a$, $b$, $\delta$ and $\Delta$, as defined in Equations (3) and (4).

When the septal invagination is formed at distance $x$ with respect to the average septum location (Fig 3B), such that $L_b^{St} = \langle L_b^{St} \rangle + x$ and $L_b^{Sw} = \langle L_b^{Sw} \rangle - x$, we then have

$$\kappa_b^{St} = \frac{1}{L_b^{St}}\frac{dL^{St}}{dt} = \frac{1}{\langle L_b^{St} \rangle + x}\alpha\varepsilon_b^{St}, \tag{10}$$

$$\kappa_b^{Sw} = \frac{1}{L_b^{Sw}}\frac{dL^{Sw}}{dt} = \frac{1}{\langle L_b^{Sw} \rangle - x}\alpha\varepsilon_b^{Sw}. \tag{11}$$

The above equations suggest a negative correlation between the growth rates of individual compartments and their respective birth lengths, which fit very well to experimental data (Fig 3C). Differential growth rate of daughter cells upon asymmetric division has been recently reported in *E. coli* (45 Preprint), where negative correlation between cell size and growth rate arises from equipartitioning of ribosomes that localize near the cell poles.

### Size-independent synthesis of growth regulators

Next, we prescribe the dynamics of $\varepsilon^{St}(t)$ and $\varepsilon^{Sw}(t)$ to predict how the cell dynamically corrects deviations in stalked-to-swarmer size ratio from their homeostatic values. If the growth regulators are synthesized in proportion to cell size such that $d\varepsilon^i/dt$ ($i$ = {St, Sw}) is proportional to $L^i$, then such a model would accelerate the growth of the bigger compartment relative to the smaller compartment, unlike what is observed in data. We therefore considered a model of size-independent synthesis such that

$$\frac{d\varepsilon^i}{dt} = k\varepsilon^i, \tag{12}$$

where $k$ is a constant rate of synthesis. Using Equations (7) and (12) we derive the time-dependence of the stalked and swarmer compartment lengths, given by,

$$L^i(t) = L_b^i + \alpha\varepsilon_b^i\left(e^{kt} - 1\right)\Big/k. \tag{13}$$

The above equation predicts that the compartments elongate at different rates unless $\gamma(t) = L^{St}(t)/L^{Sw}(t)$ is equal to the homeostatic value $\gamma^* = \varepsilon_b^{St}/\varepsilon_b^{Sw}$. In particular, the model leads to the relation

$$\frac{\kappa^{St}(t)}{\kappa^{Sw}(t)} = \frac{\gamma^*}{\gamma(t)}, \tag{14}$$

which predicts a negative correlation between the growth rate ratio and the ratio between the stalked and swarmer compartment lengths, which is in excellent agreement with experimental data (Fig 3D). As $\gamma(t)$ approaches $\gamma^*$ for $t > k^{-1}$, both the compartments grow exponentially at equal growth rates.

Combining (Equation (6)) for whole-cell elongation with Equation (12) for size-independent synthesis of growth regulators, we can derive the time evolution of cell length as $L(t) = L_b + \alpha\varepsilon_b k^{-1}\left(e^{kt} - 1\right)$. This predicts super-exponential growth of the whole cell, such that the instantaneous growth rate $\kappa(t) = L(t)^{-1}dL(t)/dt$ increases with time, in agreement with single-cell data for *C. crescentus* and *E. coli* (46, 47). Our model is thus relevant for other cell types that exhibit super-exponential growth, with appropriate modifications in the patterns of growth and cell size partitioning ratio.

### Differential growth maintains division size asymmetry in cell population

Having developed a quantitative model for asymmetric size partitioning in single cells, we asked if differential growth-mediated size correction is sufficient to achieve tight regulation of cell division ratio at the population level. To this end, we

performed stochastic single-cell simulations of growth and division for a population of asynchronous *C. crescentus* cells. Briefly, we simulated a collection of $n = 10^4$ cells where each cell consists of a stalked and a swarmer compartment that can grow at differential growth rates as given by Equation (8). At the beginning of the division cycle, we chose the position of the pre-cleavage furrow from a Gaussian distribution such that $L_b^{St}/L_b^{Sw} = 1.21 \pm 0.25$ as experimentally observed, with the constraint $L_b^{St} + L_b^{Sw} = L_b$ (Fig 1B). Initial Gaussian distribution is the only source of noise in our simulations. Based on their chosen birth lengths, the stalked and the swarmer compartments grew at size-dependent rates as deduced from experimental data (Fig 4A) and our mathematical model (Fig 4B). In our simulations, individual compartments grew with size-dependent growth rates till the crossover time $t_c = 0.6\,\tau$ (Figs 4B and S2A–D), after which they grew exponentially at constant rates independent of their size (Figs 4C and S4A–D). Interdivision time $\tau$ is computed using the formula, $\tau = \kappa^{-1}\log\left(a + \frac{\Delta}{L_b}\right)$. Once the simulations reached steady-state, we collected the data for division ratios $L_d^{St}/L_d^{Sw}$ and compared them with the experimentally obtained distribution for division ratios (Fig 4D).

Our simulation results were contrasted with two other models for growth control, where both compartments either grew at the same rate or randomly chosen growth rates independent of their initial lengths (Fig 4D). As expected, high discrepancies from experimental data were observed for non-differential growth models (same growth rates or random growth rates), whereas the empirically observed differential growth model quantitatively matched the experimentally observed distribution for cell division ratio quite well. Interestingly, the precision of cleavage positioning for differential growth model (CV = 0.06) was slightly smaller than for experimental data (CV = 0.09), suggesting additional sources of noise for division septum positioning. Taken together, the stochastic simulations based on the differential growth model show that size-dependent regulation of stalked and swarmer compartment growth is sufficient to quantitatively explain asymmetric division control and tight regulation of daughter cell size ratios in *C. crescentus* cells.

## Discussion

In this work, we studied the regulation of asymmetric cell division in *C. crescentus* using quantitative modeling and morphological analysis of single-cell data. Division site selection in *C. crescentus* is controlled by a bipolar gradient of MipZ that inhibits FtsZ polymerization (48). FtsZ assembly is localized near the mid-cell where MipZ has the lowest concentration. However, the mechanisms by which cells regulate precise positioning of the division site remain unclear. Because of concentration fluctuations in the noisy environment of a cell, divisome localization is prone to errors. There must therefore be a robust mechanism to control the precision of asymmetric size partitioning and to correct errors in size partitioning ratio.

Our results show that early in the division cycle, the growth rates of the stalked and the swarmer cell compartments can be very

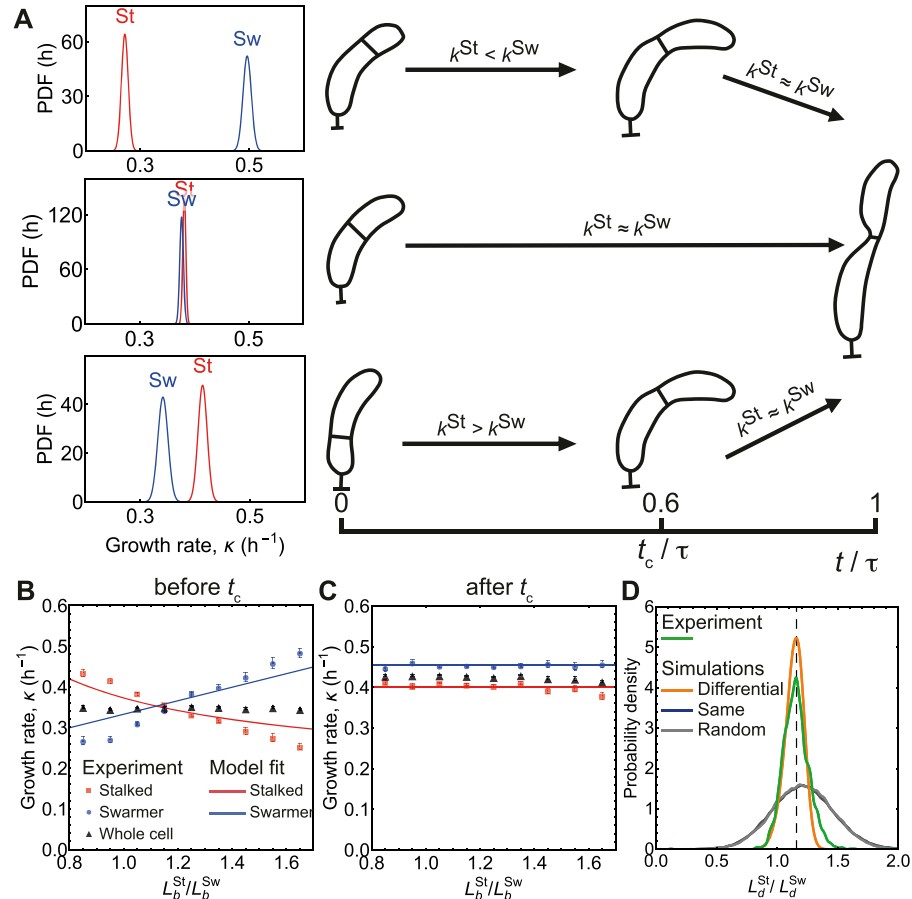

**Figure 4. Simulations of bacterial population with differential growth control predict cell length distributions in agreement with experiments.**
**(A)** Probability distribution of initial growth rates of stalked and swarmer compartments for: (top) $L_b^{St} > L_b^{Sw}$, (middle) $L_b^{St} = \gamma * L_b^{Sw}$, and (bottom) $L_b^{St} < L_b^{Sw}$. **(B)** Differential growth rates before $t_c$ are fitted to (14), yielding $\kappa^{St}(h^{-1}) = 0.186 (1 + 1/\gamma)$ and $\kappa^{Sw}(h^{-1}) = 0.166 (1 + \gamma)$, where $\gamma \equiv L_b^{St}/L_b^{Sw}$. **(C)** Growth rates after $t_c$. **(D)** Simulation predictions of division ratio statistics for different growth models. **(B, C)** When differential growth model was used (data from panel (B, C) as input), prediction of division ratio distribution (orange solid line) matched with experimental data (green solid line). CVs were 0.095, 0.068, 0.206, 0.211, and sample sizes were 2,448, 90,713, 40,310, 89,404 for experimental data, differential growth model, "same" (equal growth rate model), and random growth rate model, respectively.

different depending on the size partitioning ratio. If the stalked-to-swarmer length ratio at the beginning of the division cycle is larger than the desired size asymmetry at division, then the stalked compartment grows slower than the swarmer compartment to achieve the correct length ratio at division, and vice versa. This indicates that the insertion of peptidoglycan can be biased to either side of the division plane depending on the septum positioning. The regulation of such bias requires the cell to sense the sizes of both compartments and adjust their elongation rates accordingly.

We propose that, within an individual cell, the growth rate of each daughter cell compartment is coordinated by regulatory molecules, whose concentration decreases as the cell size increases. The concentration of the growth regulators dictates the rates for peptidoglycan production and insertion. Thus, differential growth requires concentration discrepancies for the growth regulator. Our model assuming amount-based partitioning of growth regulators is able to explain such discrepancy. It would be intriguing to test our model using the *tipN* deletion strain in *Caulobacter*, which disrupts the positioning of the division plane while leaving DNA segregation unaffected (49). To comprehensively validate our model, quantitative assessment of model predictions at various time points in mutants with division site positioning defects would be essential.

Our proposed theory is agnostic regarding the molecular identity of the growth regulators. Consequently, the molecular mechanisms underlying the spatial distribution of these regulators, ensuring equal amounts in compartments at the beginning of the division cycle, remain unknown. One possibility is that the growth regulatory molecules are bound to the genomes that are distributed in equal amounts between the two cell compartments. Chromatin-based partitioning of regulatory proteins has been recently proposed as a mechanism to control asymmetric cell division in budding yeast (20) and can serve as a robust mechanism to maintain regulatory molecules in correct proportion by limiting their diffusion. In *C. crescentus*, mRNA molecules are localized to chromosomes, which restricts their mobility (34). Because actively translating ribosomes are associated with mRNAs, they also display limited mobility and are known to set bacterial growth rate (30, 33). Thus, chromatin-based equipartitioning of translation and transcription machineries may underlie the maintenance of differential concentration of growth regulators in the stalked and the swarmer cell compartments. This prediction could be tested by immunofluorescent imaging of ribosomes and mRNA distribution across the cell at different timepoints during the cell cycle. Our proposed model for asymmetric growth control has to be contrasted with a recently proposed model for asymmetric size control in *E. coli* and *B. subtilis* (45 *Preprint*),

where ribosomes are segregated from the chromosomes and are found localized to the cell poles ([36], [37], [50]). Future experiments targeting translation and transcription machineries in different spatial compartments will elucidate the molecular origins of asymmetric cell division and size control in *C. crescentus*.

## Materials and Methods

### Acquisition of experimental data and image analysis

Single-cell data for *C. crescentus* were acquired as described in detail in reference [17], [26]. In the main text, we used the same dataset as reference [17], [26], comprising 2,448 individual cell generations at 24°C. Supplementary figures contain the data and analyses of cell shape for other temperatures. Phase-contrast images obtained were analyzed using a Python custom routine ([17]). This routine was used to determine the minimum cell width, $w_{min}$, by sampling equidistant normal lines to the cell's midline axis (Fig 1B). The length of the cell's midline axis, from the swarmer pole to the minimum cell width ($w_{min}$), defines the length of the swarmer cell compartment ($L^{Sw}$), whereas the length from $w_{min}$ to the other end of the cell defines the length of the stalked cell ($L^{St}$). Intra-generational and intergenerational dynamics of cell length and growth rate (Figs 1–3) were analyzed using custom codes written in Mathematica.

### Multivariate regression

The multivariate linear least square fits in Equations (1) and (2) were determined using RStudio. Within the dense region of the data cloud in Fig 2, single-generation data were grouped by $L_b$ and binned accordingly. These bin means were then fitted with a line using Mathematica.

## Data Availability

The data supporting the findings of this study are available from the corresponding author upon request.

## Supplementary Information

## Acknowledgements

We thank Aaron Dinner and Norbert Scherer (University of Chicago) for providing the single-cell growth and shape data for *Caulobacter crescentus* cells and Vahid Shahrezaei (Imperial College London) for sharing the data for *M. smegmatis* cells. The authors also thank Javier López-Garrido and Octavio Reyes-Matte for many useful discussions. S Banerjee acknowledges support from the Royal Society (RGF/EA/181044), the National Institutes of Health (NIH R35 GM143042), and the Shurl and Kay Curci Foundation.

## Author Contributions

TW Ng: formal analysis, investigation, visualization, methodology, and writing—original draft.
N Ojkic: conceptualization, formal analysis, validation, investigation, visualization, methodology, and writing—original draft, review, and editing.
D Serbanescu: formal analysis, validation, and investigation.
S Banerjee: conceptualization, data curation, supervision, funding acquisition, validation, investigation, visualization, methodology, project administration, and writing—original draft, review, and editing.

## Conflict of Interest Statement

The authors declare that they have no conflict of interest.

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
