## [Reviewer comments · Life Science Alliance]

Life Science Alliance

Differential growth regulates asymmetric size partitioning in *Caulobacter crescentus*

Tin Ng, Nikola Ojkic, Diana Serbanescu, and Shiladitya Banerjee

DOI: <https://doi.org/10.26508/lsa.202402591>

Corresponding author(s): *Shiladitya Banerjee, Carnegie Mellon University*

Review Timeline:

Submission Date:	2024-01-12
Editorial Decision:	2024-01-15
Revision Received:	2024-04-19
Editorial Decision:	2024-04-25
Revision Received:	2024-05-09
Accepted:	2024-05-10

Transaction Report:

Please note that the manuscript was previously reviewed at another journal and the reports were taken into account in the decision-making process at *Life Science Alliance*.

Reviewer #1 Review

Report for Author:

Its an interesting manuscript because its a new type of asymmetric growth model.

The are in particular two thing that I do not understand and that I do believe may be important to sort out

1. First I do not get what sets γ^* . I understand that it can be maintained by size independent synthesis (Eq.12) but what are the parameters that are different between the cells that makes it 1.2 and not something else?

2. In the modeling result in Fig 4D, why does the simulation reach a distribution at steady state and not a fixed value . I.e. where is the noise inputted ? It can't be the initial distribution that survives to the steady state. The system should be able to start with any reasonable distribution and reach the same steady state distribution.

Also is there some mutant phenotype that the model can explain? And what would be a good experiment to test predictions of the model? Testing a specific prediction from the model would strengthen the manuscript significantly.

Reviewer #2 Review

Report for Author:

The paper deals with the growth of *C. crescentus* cells, analyzing previously published data to study the growth of the two cellular compartments after the division plane has been determined. They make the interesting observation that the relative growth of the two daughter cells compensates for their asymmetry, the smaller compartment growing more. They build a mathematical model to capture this effect, relying on "growth factors" that determine the growth of each compartments and are partitioned more symmetrically than the volume asymmetry of the two compartments. While this is an interesting observation, the authors do not provide a molecular basis for it, not even indirect evidence using, for example, perturbing the observed phenotypes in mutants. For this reason I find the work to be a better fit for a journal such as biophysical journal rather than this journal.

Specific comments/questions:

- What is the biological significance of the compensatory mechanism described in the paper? Is there a fitness advantage? Having mutants where this is perturbed would be helpful in establishing a fitness effect, experimentally.

- Could the observations be related to the recently observed correlations of cell size and growth rate described in <https://doi.org/10.1101/2023.04.18.537336>, where the effect is attributed to the symmetric partitioning of ribosomes? Similarly, the authors relate their model to super-exponential growth - what about other bacteria where super-exponential growth is observed, such as *B. subtilis* (Nordholt et al., current biology 2020) and *E. coli* (Kar et al., elife 2021)? (Could the mechanism also be relevant for super-exponential growth in such symmetrically dividing bacteria?)

- As mentioned above, the proposed mechanism would be more compelling if there was evidence for it at the molecular level.

Reviewer #3 Review

Report for Author:

In this work, Ng et. al. investigate how asymmetrically dividing bacteria regulate cell size homeostasis. They use *C crescentus* (CC) cell as a model. CC divides asymmetrically to form stalked cell and swarmer cell with

$L^{st}/L^{sw} \sim 1.2$ at division time. At birth, L^{st}/L^{sw} shows larger variation from 1.0-1.4. Through careful quantification, the authors found that CC cells grow differentially in st and sw compartment to achieve tight $L^{st}/L^{sw} \sim 1.2$ ratio at division (i.e. high ratio CC grows faster in sw compartment and vice versa). Using a model length independent regulator model, they show how equipartition of regulator into of sw and st compartments can provide feedback necessary for shorter compartment to grow faster (and vice versa). Overall, this work shows convincing experimental evidence that there is size regulation mechanism for CC cell asymmetric division. However, the mechanism proposed is not as convincing, and the model appears to be not well constrained. Further, it is unclear if there is any clear conceptual advancement to the field of bacterial size regulation. As such, I cannot recommend publication in this journal, but I'd be happy to be convinced otherwise by the authors if they can address the following major issues.

Major points:

1. The authors claim that L^{st}/L^{sw} ratio correction is due to rate of growth of different st and sw compartment. But it is unclear to me what is the role of the placement of division plane. Does the plane move between after birth time and before division? Perhaps the entire cell is growing at some rate, but the cell division plane is also moving at some other rate to compensate for L^{st}/L^{sw} ratio. This would explain Fig. 2a for instance, as well as 'control strategy' in Eq. 3 and 4. What is known about ring formation in CC cells that would preclude this potential 'size regulation' mechanism?
2. Why does adder model predict slope of -1 for correlation of ΔL^{st} vs ΔL^{sw} ?
3. The main mechanism proposed for asymmetric size regulation is the size-independent partitioning of growth regulators. This is a hypothesis which needs further justification. Until cell division happens, st and sw compartments are connected and intracellular factors can be exchanged. The authors suggested that the growth factor is bound to the genome which is equally partitioned. But when is the genome partitioned? Does it happen before the cell division ring happens? What is the diffusion timescale of this factor? Given the length of bacteria and timescale of cell growth process, wouldn't the factors be already well mixed?
4. Related to the above point, the fits in Fig. 3C and D are tight, but the data appears to poorly constrain the fit - i.e. there are many models that can likely explain the data (e.g. a linear model). What are the confidence intervals of the parameter fits here? How tightly constrained is the model?
5. It is unclear what can we learn from the simulation results in Fig 4. The authors constructed a model, and fit the data to the model to obtain the relevant model parameter values. They then use the parameter values to run simulations and find that the simulation results fit the experimental data. Isn't this the proverbial 'getting out what you put in'? Did I miss any predictions from the simulation results, which was not put in by hand in the model?

Minor points

1. The text describing Figure 1 has many typos when referencing to the figure subpanel, which made reading and understanding the results really difficult. The authors should do their due diligence before submitting the paper.
2. Also please label the manuscript by line numbers for ease of revision.
3. "Towards the end of the division cycle, both compartments achieve similar growth rates, regardless of the initial size discrepancy (Fig. 1D)." how do we see that the growth rates are similar?
4. "C. crescentus cells exhibit a tight regulation of the division size ratio. The ratio of the stalked-to-swarmers compartment lengths at division is ≈ 1.2 " but Fig. 1C show a mean of 1??
5. "By contrast, if the growth regulators are partitioned by amounts independent of size, then $k_i \propto 1/L_i$." If this is independent of size, why is there still a L dependence?
6. Some terminology: it is unclear why the authors call Eq 3 and Eq 4 a 'control strategy'. Isn't this just the phenomenology of the system?

January 15, 2024

Re: Life Science Alliance manuscript #LSA-2024-02591-T

Prof Shiladitya Banerjee
Carnegie Mellon University
Physics
Pittsburgh

Dear Dr. Banerjee,

Thank you for submitting your manuscript entitled "Differential growth drives cell size homeostasis in asymmetrically dividing bacteria" to Life Science Alliance. We invite you to submit a revised manuscript addressing the following Reviewer comments:

- Address Reviewer 1's comments.
- Address Reviewer 2's Specific comment #2 via added Discussion.
- Address Reviewer 3's Major points #1, 2, 4 & 5, and the Minor points.

Thank you for this interesting contribution to Life Science Alliance. We are looking forward to receiving your revised manuscript.

Sincerely,

Eric Sawey, PhD
Executive Editor
Life Science Alliance
<http://www.lsa-journal.org>

- A letter addressing the reviewers' comments point by point.
- An editable version of the final text (.DOC or .DOCX) is needed for copyediting (no PDFs).
- High-resolution figure, supplementary figure and video files uploaded as individual files: See our detailed guidelines for preparing your production-ready images, <https://www.life-science-alliance.org/authors>
- Summary blurb (enter in submission system): A short text summarizing in a single sentence the study (max. 200 characters including spaces). This text is used in conjunction with the titles of papers, hence should be informative and complementary to the title and running title. It should describe the context and significance of the findings for a general readership; it should be written in the present tense and refer to the work in the third person. Author names should not be mentioned.
- By submitting a revision, you attest that you are aware of our payment policies found here: <https://www.life-science-alliance.org/copyright-license-fee>

B. MANUSCRIPT ORGANIZATION AND FORMATTING:

Editor Comments:

Thank you for submitting your manuscript entitled "Differential growth drives cell size homeostasis in asymmetrically dividing bacteria" to Life Science Alliance. We invite you to submit a revised manuscript addressing the following Reviewer comments:

- Address Reviewer 1's comments.
- Address Reviewer 2's Specific comment #2 via added Discussion.
- Address Reviewer 3's Major points #1, 2, 4 & 5, and the Minor points

We thank the editor for inviting a submission of our manuscript in Life Science Alliance. Below we address the specific reviewer comments as asked by the Editor. To more clearly reflect our central findings, we have modified the title of the manuscript to "Differential growth regulates asymmetric size partitioning in *Caulobacter crescentus*"

Reviewer #1:

Its an interesting manuscript because its a new type of asymmetric growth model.

There are in particular two thing that I do not understand and that I do believe may be important to sort out

1. First I do not get what sets γ^* . I understand that it can be maintained by size independent synthesis (Eq.12) but what are the parameters that are different between the cells that makes it 1.2 and not something else?

The parameter γ^* is the average size ratio between stalked to swarmer cells at birth, as defined in Eq. 9. It is dependent on the parameters that regulate the size of stalked and swarmer compartments, namely the parameters a , b , δ , and Δ , as defined in Eqs. 3-4. We have clarified this point in the revised manuscript.

2. In the modeling result in Fig 4D, why does the simulation reach a distribution at steady state and not a fixed value. i.e. where is the noise inputted? It can't be the initial distribution that survives to the steady state. The system should be able to start with any reasonable distribution and reach the same steady state distribution.

As experimentally observed, during steady-state growth, the size ratio of stalked to swarmer compartments (L_{st}/L_{sw}) at birth follows an approximate Gaussian distribution with mean 1.21 and standard deviation 0.25 (lines 402-404 of the main text). This initial size ratio distribution is an input to the model, and is the only source of noise in our population-scale simulations.

Fig 4D shows different steady-state distribution as they are obtained from different models. Based on the initial value of L_{st}/L_{sw} , growth rates are assigned for stalked and swarmer compartments using the

various models presented in Fig 4D. (Random – growth rates for both compartments are chosen randomly; Differential – growth rates are assigned as observed from Fig. 3B; Same – growth rates are the same for both compartments). The initial distribution of Lst/Lsw at birth is solely a contributor to the distribution of Lst/Lsw at the division.

We now clarify the source of noise in the simulation paragraph of the main text (lines 402-404).

Also is there some mutant phenotype that the model can explain? And what would be a good experiment to test predictions of the model? Testing a specific prediction from the model would strengthen the manuscript significantly.

Unfortunately, we do not have access to single-cell microfluidics data on relevant *Caulobacter* mutants, as such experiments have not been conducted. We cannot perform those experiments ourselves as we are a purely theory/modeling group. Nonetheless, it would be intriguing to test our model using the tipN deletion strain in *Caulobacter* (Lam et al., Cell 2006), which disrupts the placement of the division plane while leaving DNA segregation unaffected. To comprehensively validate our model, quantitative assessment of model predictions at various time points in mutants with division site positioning defects would be essential. We have added this point to the discussion section.

Reviewer #2:

Specific Comment #2: Could the observations be related to the recently observed correlations of cell size and growth rate described in <https://doi.org/10.1101/2023.04.18.537336>, where the effect is attributed to the symmetric partitioning of ribosomes?

We thank the reviewer for pointing out the preprint by van Heerden et al (bioRxiv, 2023), where the authors report a similar negative correlation between cell size and growth rate in *E. coli*. We have cited this paper now. However, the proposed mechanism for negative correlation between cell size and growth rate is different since it is attributed to the inhomogeneous distribution of ribosomes. In particular, *E. coli* cells localize their ribosomes to the cell poles, whereas *C. crescentus* ribosomes are homogeneously distributed (Montero-Llopis P et al., Biophys. J. 2012). We comment on this in the revised manuscript.

Similarly, the authors relate their model to super-exponential growth - what about other bacteria where super-exponential growth is observed, such as *B. subtilis* (Nordholt et al., current biology 2020) and *E. coli* (Kar et al., elife 2021)? (Could the mechanism also be relevant for super-exponential growth in such symmetrically dividing bacteria?)

As shown by us recently, super-exponential growth arises from autocatalytic production of ribosomes, whose concentration determine growth rate (Cylke et al Biophys J 2023). In our model, the autocatalytic growth regulators play a similar role as actively translating ribosomes, since growth rate is proportional

to the concentration of the growth regulators. Our model is thus applicable to other super-exponentially growing bacteria such as *E. coli* with appropriate modifications, such as the distribution of growth pattern and cell size partitioning ratio. We comment on this in the revised manuscript (lines 378-386). Note that *B. subtilis* does not strictly exhibit super-exponential growth, since the growth rate is non-monotonic, showing phases of super-exponential and sub-exponential growth (Nordholt et al., current biology 2020). This may arise from temporal changes in growth pattern, which is beyond the scope of this paper.

Reviewer #3:

In this work, Ng et. al. investigate how asymmetrically dividing bacteria regulate cell size homeostasis. They use *C. crescentus* (CC) cell as a model. CC divides asymmetrically to form stalked cell and swarmer cell with $L^{st}/L^{sw} \sim 1.2$ at division time. At birth, L^{st}/L^{sw} shows larger variation from 1.0-1.4. Through careful quantification, the authors found that CC cells grow differentially in st and sw compartment to achieve tight $L^{st}/L^{sw} \sim 1.2$ ratio at division (i.e. high ratio CC grows faster in sw compartment and vice versa). Using a model length independent regulator model, they show how equipartition of regulator into of sw and st compartments can provide feedback necessary for shorter compartment to grow faster (and vice versa). Overall, this work shows convincing experimental evidence that there is size regulation mechanism for CC cell asymmetric division. However, the mechanism proposed is not as convincing, and the model appears to be not well constrained. Further, it is unclear if there is any clear conceptual advancement to the field of bacterial size regulation. As such, I cannot recommend publication in this journal, but I'd be happy to be convinced otherwise by the authors if they can address the following major issues.

Major points:

1. The authors claim that L^{st}/L^{sw} ratio correction is due to rate of growth of different st and sw compartment. But it is unclear to me what is the role of the placement of division plane. Does the plane move between after birth time and before division? Perhaps the entire cell is growing at some rate, but the cell division plane is also moving at some other rate to compensate for L^{st}/L^{sw} ratio. This would explain Fig. 2a for instance, as well as 'control strategy' in Eq. 3 and 4. What is known about ring formation in CC cells that would preclude this potential 'size regulation' mechanism?

This is an excellent point raised by the reviewer. From the data we can only conclude that the stalked and the swarmer compartments elongate at different rates, but it doesn't preclude a model where the division plane may be moving relative to the cell, in order to compensate for the L^{st}/L^{sw} ratio. We have discussed this possibility in the Results section before introducing the model.

However, note that timelapse of FtsZ-GFP doesn't seem to indicate relative motion of the division ring (Lambert et al, iScience 2018), and any mechanism for the translocation of the Z-ring is unknown at the moment. In *Caulobacter* cells, division ring placement occurs right at the beginning of the stalked cell cycle, which is where cells would eventually divide (Govers and Jacobs-Wagner, Current Biology 2020).

2. Why does adder model predict slope of -1 for correlation of ΔL^{st} vs ΔL^{sw} ?

A negative slope of -1 for the correlation of ΔL^{st} vs ΔL^{sw} comes directly from the definition of the added length: $\Delta L = \Delta L^{st} + \Delta L^{sw}$. Therefore $\Delta L^{st} = \Delta L - \Delta L^{sw}$, where ΔL is the total added length. Since ΔL is constant for the adder model, we would expect a slope of -1 for the correlation of ΔL^{st} vs ΔL^{sw} . In the text and the caption of Fig. 2A, the expected negative correlation is clarified.

3. The main mechanism proposed for asymmetric size regulation is the size-independent partitioning of growth regulators. This is a hypothesis which needs further justification. Until cell division happens, st and sw compartments are connected and intracellular factors can be exchanged. The authors suggested that the growth factor is bound to the genome which is equally partitioned. But when is the genome partitioned? Does it happen before the cell division ring happens? What is the diffusion timescale of this factor? Given the length of bacteria and timescale of cell growth process, wouldn't the factors be already well mixed?

Although addressing this comment isn't necessary for LSA, we wanted to make it clear that DNA replication occurs very early on in the cell cycle (Collier, Plasmid 2012), right at the swarmer-to-stalked cell transition, and before FtsZ ring forms. The growth factor can be assumed to be ribosomes, since ribosome concentration regulates growth rate of bacteria (Scott et al 2010). It has been observed that actively translating ribosomes (which constitute 70% of the ribosomes) are bound to the genome and do not diffuse much, with an extremely low diffusion coefficient $\sim 0.0002 \mu\text{m}^2/\text{s}$ (Llopis et al Biophys J 2012). Free ribosomes are also slow diffusion with a diffusion coefficient $\sim 0.018 \mu\text{m}^2/\text{s}$. We have made these points clearer in the main text.

4. Related to the above point, the fits in Fig. 3C and D are tight, but the data appears to poorly constrain the fit - i.e. there are many models that can likely explain the data (e.g. a linear model). What are the confidence intervals of the parameter fits here? How tightly constrained is the model?

In Fig. 3C and D, the nonlinear model in Eq. 8 is fitted to the binned data that are shown with black dots. The model has only one free parameter that has been fitted. In Fig. 3D and the graph below, the blue line represents the best model fit to the binned data, with fitted parameters and standard error given as 1.26 ± 0.03 . The nonlinear model fit for all experimental points (dashed black line below) provides a similar fitting model parameter of 1.286 ± 0.012 . Below, the green lines represent mean prediction bands, and the orange lines represent a single-point prediction band for 95% confidence intervals. To account for the full spread of experimental data, without binning, a 95% confidence interval will require the model parameter to be in the range between 1.26 - 1.31.

5. It is unclear what can we learn from the simulation results in Fig 4. The authors constructed a model, and fit the data to the model to obtain the relevant model parameter values. They then use the parameter values to run simulations and find that the simulation results fit the experimental data. Isn't this the proverbial 'getting out what you put in'? Did I miss any predictions from the simulation results, which was not put in by hand in the model?

The population-scale simulations in Fig. 4 aim to assess the single-cell level for differential growth is adequate to account for the distribution of cell size partitioning ratios within a bacterial population. Our findings demonstrate that the differential growth model effectively captures the spread of the length ratio distribution without the need to incorporate various sources of noise stemming from stochastic growth and division processes. In our simulations, the only source of noise considered was the initial distribution of lengths at the onset of the cell cycle, which was determined from experimental data. Furthermore, the simulations demonstrate that alternative models of growth control, such as random or uniform growth rates across compartments regardless of their initial length ratio, disrupt the tight regulation of length ratio at division, resulting in a significantly broader distribution.

Minor points

1. The text describing Figure 1 has many typos when referencing to the figure subpanel, which made reading and understanding the results really difficult. The authors should do their due diligence before submitting the paper.

The typos are now corrected.

2. Also please label the manuscript by line numbers for ease of revision.

Line numbers have now been included.

3. "Towards the end of the division cycle, both compartments achieve similar growth rates, regardless of the initial size discrepancy (Fig. 1D)." how do we see that the growth rates are similar?

We thank the reviewer for pointing this out. Indeed, it is not obvious from Fig 1D that both compartments achieve similar growth rates. The reference should have been to Fig 1E and 1G, where we clearly see that stalked and swarmer grow at similar rates after t_c . We now point the readers to panels 1E and 1G.

4. "C. crescentus cells exhibit a tight regulation of the division size ratio. The ratio of the stalked-to-swarmer compartment lengths at division is ≈ 1.2 " but Fig. 1C show a mean of 1??

In Fig. 1C we plotted the distributions of L_{st}/L_{sw} normalized by their mean values at different time points. That is the mean appears to be 1 in Fig 1C. At each time point, L_{st}/L_{sw} was normalized by the corresponding mean value so that the distributions were aligned. Therefore, the readers can easily observe the spread of the distributions that we quantified by CV. To help the reader, we now explain the normalization procedure in the caption of Fig. 1C.

5. "By contrast, if the growth regulators are partitioned by amounts independent of size, then $k_i \propto 1/L_i$." If this is independent of size, why is there still a L dependence?

Even though the amounts of growth regulators are partitioned independent of compartment size, their concentrations (amount/size) would still be dependent on size. According to our model Eq. 8, growth rate of each compartment (k_i) scales with the concentration of growth regulators in that compartment. Thus k_i would be proportional to $1/L_i$. We have clarified this point in the revised manuscript.

6. Some terminology: it is unclear why the authors call Eq 3 and Eq 4 a 'control strategy'. Isn't this just the phenomenology of the system?

It is true that Eq. 3 and 4 are phenomenological models derived from data. However, they also represent strategies for size control, as eq. 3-4 relate cell compartment size at birth to cell compartment size division. To remove any confusion, we have removed the term 'strategy'.

April 25, 2024

RE: Life Science Alliance Manuscript #LSA-2024-02591-TR

Prof. Shiladitya Banerjee
Carnegie Mellon University
Physics
5000 Forbes Ave
Pittsburgh 15213

Dear Dr. Banerjee,

Thank you for submitting your revised manuscript entitled "Differential growth regulates asymmetric size partitioning in *Caulobacter crescentus*". We would be happy to publish your paper in Life Science Alliance pending final revisions necessary to meet our formatting guidelines.

- please be sure that the authorship listing and order is correct
- please upload your main manuscript text as an editable doc file
- please add ORCID ID for the corresponding author -- you should have received instructions on how to do so
- please add the Twitter handle of your host institute/organization as well as your own or/and one of the authors in our system
- please remove figures from the manuscript file and leave them uploaded separately only
- please add your main and supplementary figure legends to the main manuscript text after the references section
- please add a conflict of interest statement to your main manuscript text
- please add an Author Contributions section to your main manuscript text
- please add callouts for Figures 4A; S1A-C and S3A-D to your main manuscript text

A. FINAL FILES:

B. MANUSCRIPT ORGANIZATION AND FORMATTING:

Thank you for your attention to these final processing requirements. Please revise and format the manuscript and upload materials within 4 days.

Sincerely,

May 10, 2024

RE: Life Science Alliance Manuscript #LSA-2024-02591-TRR

Prof. Shiladitya Banerjee
Carnegie Mellon University
Physics
5000 Forbes Ave
Pittsburgh 15213

Dear Dr. Banerjee,

Thank you for submitting your Research Article entitled "Differential growth regulates asymmetric size partitioning in *Caulobacter crescentus*". It is a pleasure to let you know that your manuscript is now accepted for publication in Life Science Alliance. Congratulations on this interesting work.

DISTRIBUTION OF MATERIALS:

Again, congratulations on a very nice paper. I hope you found the review process to be constructive and are pleased with how the manuscript was handled editorially. We look forward to future exciting submissions from your lab.

Sincerely,
